# Delaying the Stall of A Low-Wing Aircraft Using A Novel Powerful Vortex Generator

**Mostafa Goharshadi**  **and Masoud Mirzaei ***

Department of Aerospace Engineering, K. N. Toosi University of Technology, Tehran 19967-15433, Iran
* Correspondence: mirzaei@kntu.ac.ir

**Abstract:** In this study, a new aerodynamic surface concept is introduced, which is a powerful vortex generator (PVG). It can delay the stall point in a low-wing aircraft. This delay leads to a significant increase in the $C_{Lmax}$ of an aircraft. The results of this research show that the use of PVG, due to its longitudinal position, does not affect the aerodynamic center of the aircraft as well as its static stability. This is an advantage for this method compared to the method based on LEX, in which the aerodynamic center moves forward and the static stability of the aircraft reduces. As a case study, this research focused on a low-wing advanced training jet. Additionally, the aerodynamic characteristics of the aircraft were investigated in three points, including takeoff /landing condition, one maneuvering point, and one MMO condition. To evaluate the concept of PVG in more realistic situations, the wing airfoil was optimized at the same three points using the adjoint method. Then, the effect of PVG on various configurations of the aircraft, including the clean configuration and the different types of flap, was investigated. Since all the analyses were performed using computational fluid dynamics, at first, the validation of numerical methods was conducted on two test cases in low-speed and high-speed flows. The results of the case study show that the PVG greatly delays the separation and increases the value of $C_{Lmax}$. For example, in the case of a hinged leading-edge flap and single slotted trailing-edge flap, more than 12 degrees of delay in the stall was achieved, and the value of $C_{Lmax}$ increased from 1.4 to 2.05 (46% increase).

**Keywords:** vortex generator; stall; adjoint; low-wing; aircraft

## 1. Introduction

Delaying the aircraft stall to achieve higher $C_{Lmax}$ (maximum lift coefficient) has been of much interest in using various types of boundary-layer control (BLC) in recent decades. As well as minimizing air vehicle drag by delaying the transition from laminar to turbulent flow, BLCs are used to delay the flow separation of high angles of attack lifting sur-faces that experience a strong adverse pressure gradient.

There are three types of BLC including active, semi-active, and passive. The active BLC includes blowing/suction systems, morphing surface actuators, plasma actuators, etc. In blowing/suction systems, low-energy layers are sucked in, or high-energy flows are blown [1]. Plasma actuators are an example of this type of BLC. In these actuators, the induced velocity energizes the low-energy boundary layers [2]. In morphing surface actuators, piezoelectric actuators excite the turbulent boundary layer. One of the biggest disadvantages of active methods is the need for a secondary source of energy. In addition, active flow control generally imposes additional weight and maintenance costs [3]. There are numerous review articles to find out the pros and cons of different types of active BLC (e.g., [1–3]). The semi-active BLC includes a movable type of flap and slat. These BLCs are active at some points of the flight envelope, such as takeoff, landing, and sometimes maneuver points, and will be inactive at other points of the flight envelope.

The last type of lifting surface BLC is the passive method. One way of the passive BLC method is to optimize the shape of the lifting surface to delay the flow separation.

Other methods use fixed-component high-lift devices, which are classified into two types depending on their location, either in the boundary layer region or outside. Figure 1 shows lifting surface BLC categories.

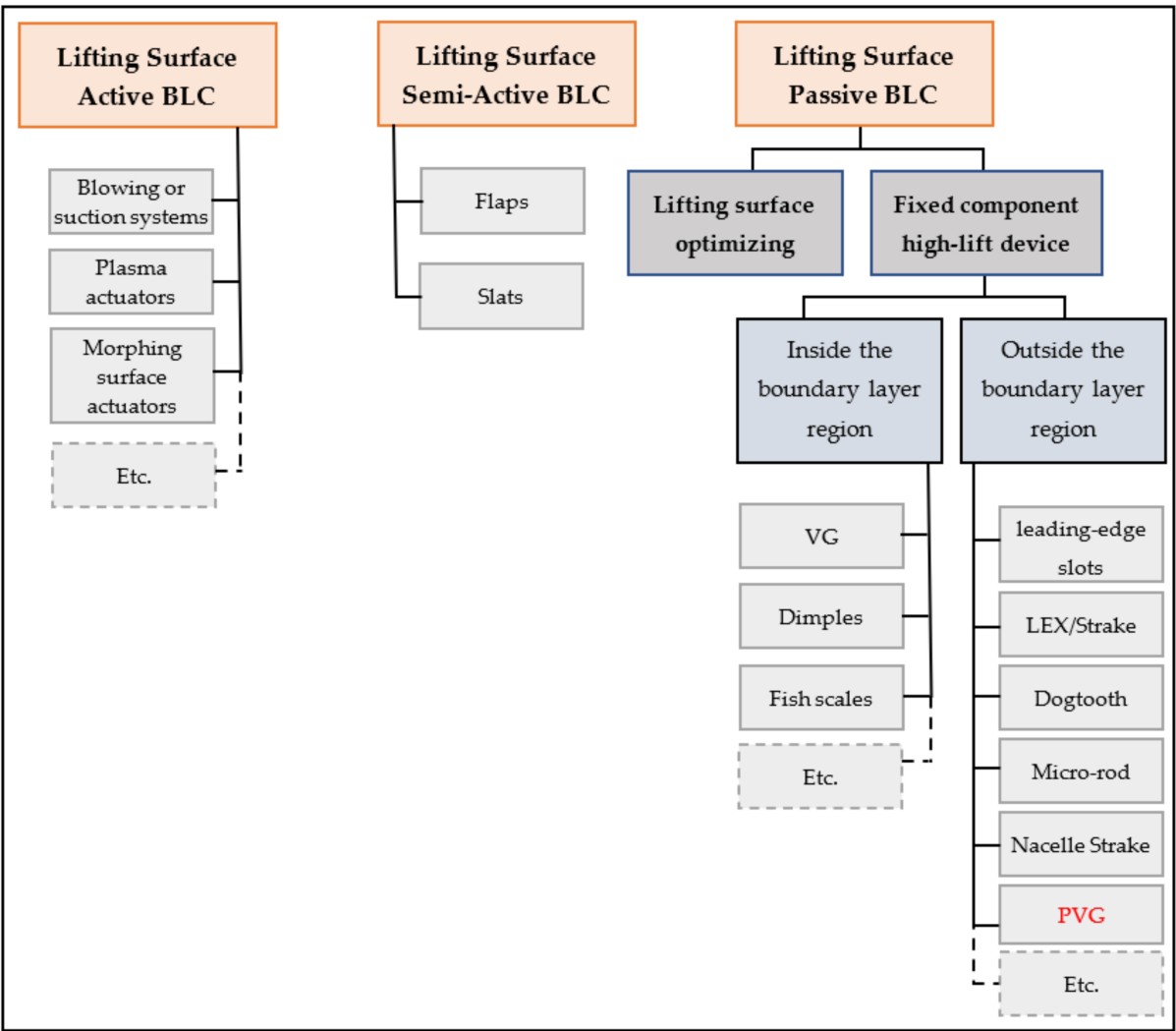

**Figure 1.** The categories of lifting surface BLC.

To compare any lifting surface BLC or investigate a new idea, it is necessary to examine all effects in a multidisciplinary approach. Figure 2 depicts various fields necessary to include in a comprehensive review. These fields all originate from the mission and flight envelope of an aircraft. They can include aerodynamics, performance, stability and control, air loading, structure, system weight imposed on the aircraft, system reliability, and maintainability, etc.

In the aerodynamic analysis of a high-lift device, in addition, to achieve an increase in $C_{Lmax}$ at low speeds during takeoff and landing, the aircraft drag change at high speeds at cruise, maneuver, MMO, etc., points is also important. In addition, a change in the $C_{m0}$ (pitching moment coefficient at zero angle of attack) of an aircraft can change the balanced loading and also impose a trim drag. Furthermore, it can affect the air loading of the wing and rear body as well as the controllability of the aircraft. In addition, a change in the $C_m - \alpha$ (pitching moment coefficient versus angle of attack) slope can change the aerodynamic center of the aircraft and thus its stability. Additionally, the weight of the system imposed on the aircraft should be reasonable, along with increasing the $C_{Lmax}$.

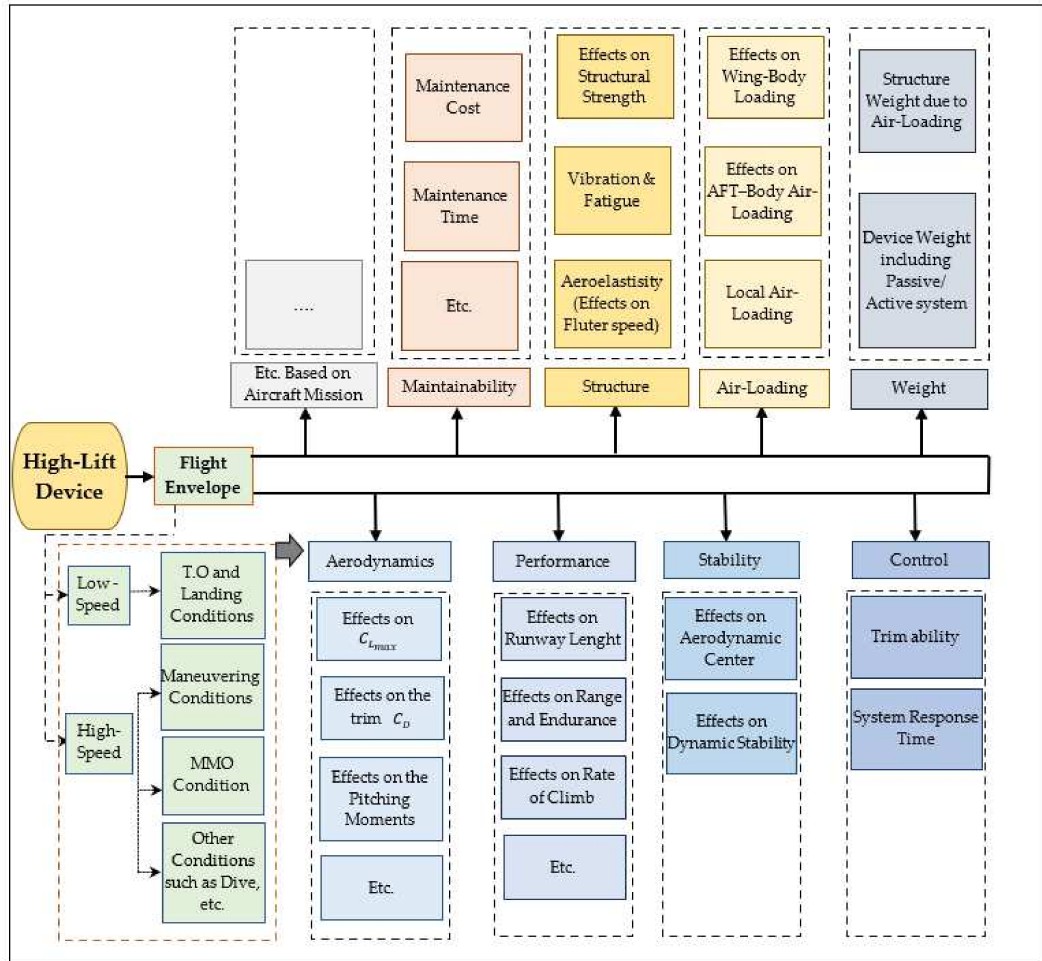

**Figure 2.** A typical multidisciplinary approach of a high-lift device.

During some research on designing an advanced low-wing training jet, the authors came up with an innovative design to delay the aircraft stall. This is a powerful vortex generator, which is attached to the body and is located at the top of the wing (Figure 2). PVG is a fixed component high-lift device and, unlike small conventional vortex generators, locates outside the boundary layer region. The performance and advantages of this new aerodynamic surface compared to other high-lift devices are discussed in this article. According to the location of PVG, this device can be used in combination with other BLC methods (except Leading-Edge Extension (LEX)). In this research, the effect of PVG along with semi-active BLC, i.e., several types of leading-edge and trailing-edge flaps, are investigated. Additionally, the combination of PVG with the lifting surface optimization method is also presented at the same time. To evaluate the concept of PVG in a more realistic manner, the wing airfoil is optimized in three-point optimization, including takeoff condition, maneuver point, and cruise condition, using the adjoint method.

The adjoint method has been well-developed in recent decades [4–6]. The computational cost of this method is independent of the number of design variables. Hence, the low computational cost is an advantage of this method over other gradient-based methods. This method involves two different approaches, including continuous and discrete. The second approach results in more precise gradients [5], so it has been implemented in this work. To accelerate the optimization process, a 2D all-speed flow and adjoint solver based on a viscous unstructured finite volume method is developed [7,8]. To do this, the 3D conditions of the aircraft are reduced to 2D conditions using conventional methods.

As stated before, it is possible to use PVG together with other high-lift devices. However, in the case of LEX, it is not reasonable to use it at the same time. LEX, an aerodynamic

surface, has been proposed to delay the aircraft stall [9–11]. Like PVG, it prevents flow separation by creating a vortex flow over a wide area of the wing. Therefore, there is a possibility of interference between these vortex flows. Herein, it is necessary to make a comparison between the performance of LEX and PVG.

Generally, the position of a LEX in the front region of a wing moves forward the aerodynamic center, and therefore, the static stability of an aircraft decreases. As an example, the wind tunnel tests of the F-16 model show that installing a LEX shifts the aerodynamic center from approximately 7.2 to 3 percent of $\bar{c}$ (mean aerodynamic chord) in the linear region of the lift curve [11]. To compensate for the LEX effect, either the aircraft gravity center must move forward, or an artificial stability system (such as stability augmentation system (SAS) or fly-by-wire (FBW)) should be used. The requirements impose complexity and cost. As it is shown in the results, PVG does not move the aerodynamic center of the aircraft, and as a result, the static stability of the aircraft does not change. In contrast to this advantage, PVG in this research is proposed for installation on low-wing aircraft, which is a limitation compared to LEX. For the use of PVG in high-wing aircraft, other innovative designs are required, which are beyond the scope of this work.

Other methods, such as using dogtooth or vortex generators (VGs), delay the separation in the local area of the wing and have local effects. LEX, dogtooth, and VGs are sometimes used together to delay separation in the fuller part of the wing.

VGs alone have been investigated in several previous reports, and good review articles have been published in this field [1]. In a three-element high-lift airfoil at M = 0.2 and Re = $5 \times 10^6$, 10% lift increase and in RAE 5243 transonic airfoil at M = 0.67 and Re = $19 \times 10^6$, 20% lift increase were reported [1]. It should be emphasized that these data are for the airfoil alone, and the effectiveness of the results will be reduced in an aircraft in the presence of a fuselage. However, it seems that the combination of PVG with conventional VGs leads to synergy in the results, which is another research.

Another device in the passive control of the boundary layer is the nacelle strake. This device compensates for the negative effect of turbofan engine nacelles on the wing flow field. It increases the $C_{Lmax}$ value by about 5.5% in the high lift configuration [12]. This device is installed in the front of a wing, and its functionality is similar to conventional strake (LEX). The position, size, and effects of this device are significantly different from the proposed PVG device shown in the results.

In addition to the above methods, other passive techniques such as dimples [13–15], fish scales [15], and micro-rod [16] have been introduced to delay the flow separation in recent years. As long as these different techniques are not investigated in a single platform or practically used in an operational aircraft, it is not possible to state accurately their advantages and disadvantages. In the dimple and fish scale devices, different results have been reported. The differences can be due to their shape, position, size, and arrangement, as well as the analysis conditions. To examine them completely, according to Figure 2, it is necessary to consider various issues such as their effect on the pitching moment, control and stability, buffeting, the strength of the structure, etc. Furthermore, the effect of combining the use of dimple and fish scales with PVG on aerodynamic coefficients needs an additional study. In the micro-rod device, it prevents severe stalling and creates a delay of about 2 degrees in the stall point [16]. Again, it seems that in the practical use of this device, as well as the leading-edge slot device, one should pay attention to its effects on the aircraft cruise condition and structure vibration.

In this work, the introduced method was examined by itself. Only a general comparison is made with the LEX and Nacelle strake techniques as methods that have been applied in various aircrafts.

In the results section, PVG is implemented on a sample of an advanced low-wing training jet. In order to reduce the complexity and the volume of computing mesh, the results are provided for the wing-body configuration.

According to Figure 2, in presenting PVG as a new fixed component high-lift device, extensive issues should be checked. In the first step, in addition to the main issue of

increasing the $C_{Lmax}$ of the aircraft in takeoff/landing conditions at low-speed flows, the effects of PVG on aerodynamic coefficients are investigated in two conditions of high-speed flows, namely MMO and a maneuvering point. In the next step, according to Figure 1, the combination of PVG as a fixed component high-lift device with the method of lifting surface optimization is conducted in order to achieve better results. For this purpose, using the adjoint method, the wing airfoil is optimized in the same three points of takeoff/ landing condition, MMO condition, and the maneuvering point. To check the optimization effects in the presence of PVG, a comparison between results in the same three points is presented. In the final step, again according to Figure 1, the combination of PVG as a passive BLC with a semi-passive BLC is investigated. In other words, the effect of PVG on several combinations of the leading-edge flap and trailing-edge flap is investigated. Moreover, a conceptual consideration of the other issues raised in Figure 2 is conducted.

To validate the 3D numerical results, two CFD (computational fluid dynamic) test cases have been selected at low-speed (DLR-F11) [17] and high-speed (AGARD B) [18–20].

## 2. The Considered Case: A Low-Wing Advanced Training Jet

As an example, the wing-body configuration of an advanced low-wing training jet is generally designed to explore the new concept. The geometry of the configuration is somewhat similar to jet trainers such as AT-3 and Yasin (Iranian jet trainer), but there is no one-to-one similarity. In other words, the new concept can be applied to such aircrafts.

The reference values of the investigated configuration are wing area = 23.6 m$^2$, mean aerodynamic chord ($\bar{c}$) = 2.4 m, moment reference point = 25% $\bar{c}$ and its airfoil is the Modified SC(2)-410.

## 3. Airfoil Optimization

Airfoil optimization of the aircraft was assumed based on three design points including: (1) as much as possible decreasing the stall speed, (2) passing a maneuvering point with maximum load factor, (3) minimizing drag in MMO point. Therefore, the total target function of three-point optimization, I, is defined in Equation (1).

$$I = w_1\ I_1 + w_2\ I_2 + w_3\ I_3 \tag{1}$$

In Equation (1), $I_1$–$I_3$ are the target function of each point of optimization and $w_1$–$w_3$ are their weighting coefficients. The weighting coefficients were chosen as 0.5, 0.3, and 0.2, respectively. Additionally, a maneuver point of 18,000 feet and Mach 0.6 and an MMO point of 30,000 feet and Mach 0.87 were selected as the conditions of optimization points. All these choices help to evaluate the aircraft in more realistic situations and can be changed by the aircraft designer based on an aircraft mission. Due to the high subsonic speed of the aircraft, the SC(2)-410 supercritical airfoil was selected as the initial airfoil with some modifications. This airfoil was then optimized based on the adjoint method at the mentioned design. The formulations and details of the method are out of the scope of the article [7,8]. Figure 3 shows the initial and the optimized SC(2)-410 airfoils together with the original SC(2)-410 airfoil.

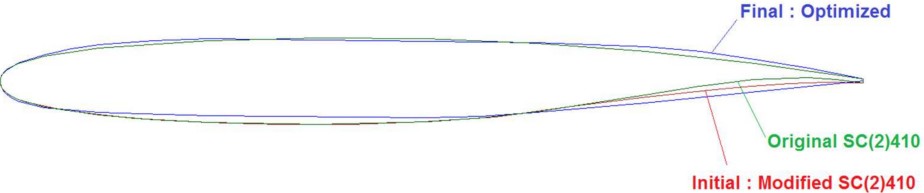

**Figure 3.** The initial airfoil versus the final optimized airfoil.

## 4. PVG; A Powerful Vortex Generator

Figure 4 shows an innovative aerodynamic surface called PVG. This aerodynamic surface, designed for low-wing aircraft, is located adjacent to the body and above the root of

the wing. Figure 5 shows how the vortex generated by the PVG can delay wing separation.

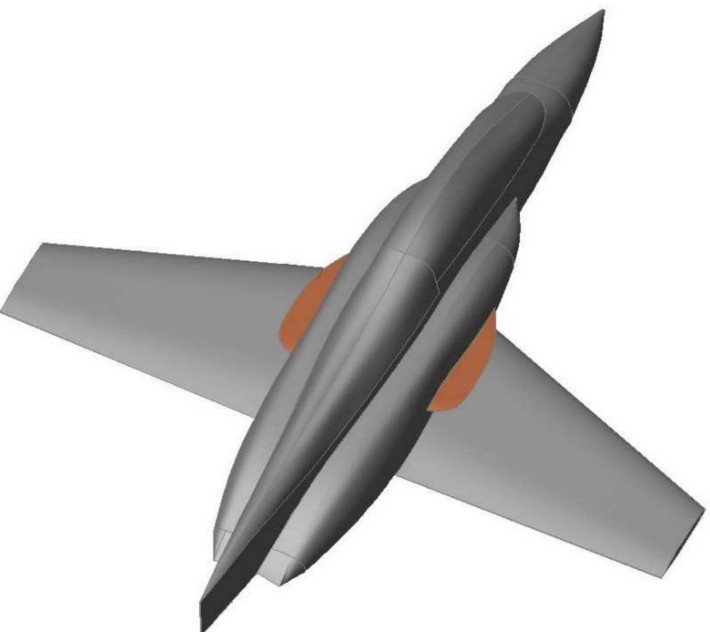

**Figure 4.** PVG, a powerful vortex generator.

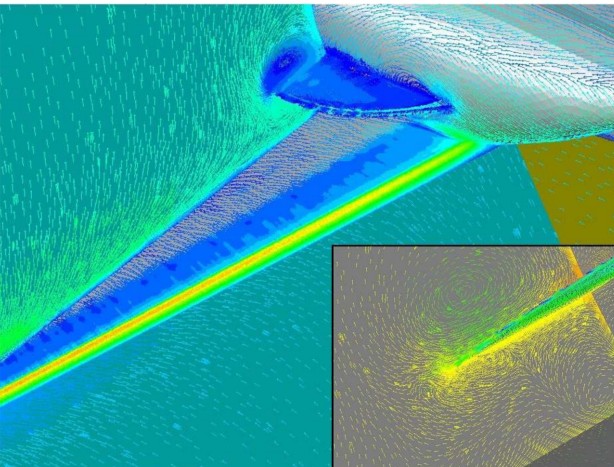

**Figure 5.** The vortex generated due to PVG.

The position and the shape of PVG were obtained in a trial-and-error procedure. In this article, the PVG effects on aerodynamic coefficients are considered. The position and the shape of PVG can be presented in another article in wide complementary investigations.

## 5. CFD Validation

To ensure the results of numerical methods, two issues of verification and validation (V&V) are needed. In verification, two subjects, including numerical code verification and solution verification, are considered, and in validation, the degree of agreement between the numerical model and reality is determined [21]. In this work, the well-known software FLUENT, which is used in 3D computational fluid dynamics (CFD) analysis, ensures the first part of verification, i.e., code verification. In order to ensure the second part of verification, i.e., verifying the solution, as well as the validating CFD, two experimental models were selected in low-speed and high-speed flows for which valid wind tunnel results are available. These include the DLR-F11 wing-body model in low-speed flow and the AGARD-B model in high-speed flow.

### 5.1. Low-Speed Test Case: DLR-F11

DLR-F11 is a wind tunnel model widely used as a validation of CFD results [17]. The model is shown in Figure 6. Herein, the low-speed results at Re = 15.1 million were used. A viscous incompressible flow solver with SST k-$\omega$ turbulence model and second-order discretization is selected in CFD simulation. The generated mesh on the DLR-F11 model is unstructured with 10-layer prism cells (the growth rate 1.2) to capture the boundary layer in the vicinity of the walls. It includes about 9 million computational cells based on half of the computational domain.

The comparison of CFD and wind tunnel results is presented in Figure 6a. In the $C_L - \alpha$ diagram, there is a good agreement between CFD and experimental results, so that the values of $C_{Lmax}$ are close to each other, and the prediction of the stall area is almost well performed. In the drag coefficient versus angle of attack diagram ($C_D - \alpha$), there is an excellent agreement at low angles of attack and a good trend prediction at higher angles. Additionally, the good trend prediction is approximately shown in $C_m - \alpha$ diagram.

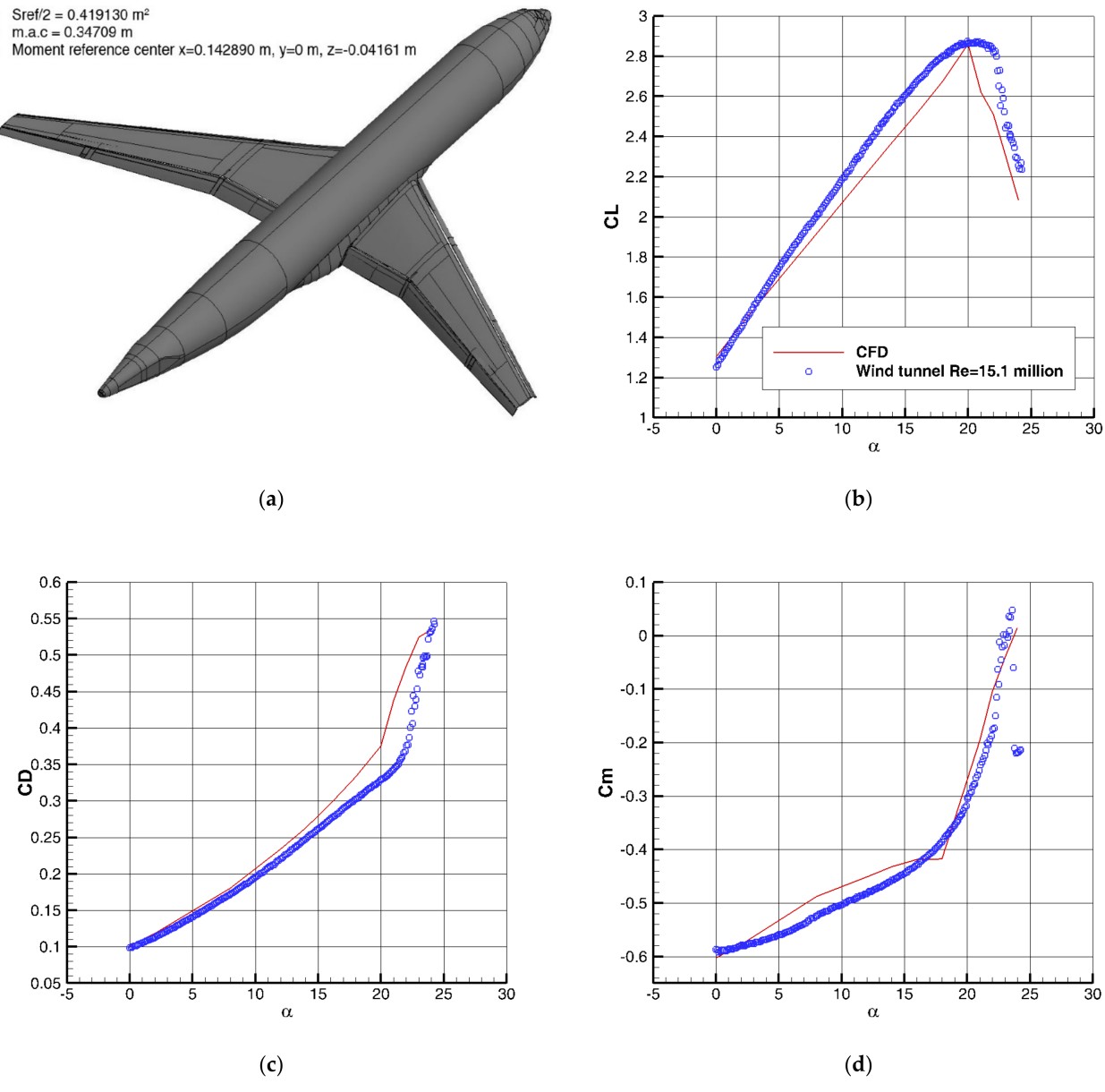

**Figure 6.** Low-speed test case CFD validation: (**a**) DLR-F11 model; (**b**) $C_L - \alpha$ diagram; (**c**) $C_D - \alpha$ diagram; (**d**) $C_m - \alpha$ diagram.

### 5.2. High-Speed Test Case: AGARD-B

AGARD-B is a standard model which is widely used as a wind tunnel calibration or CFD validation [18–20]. The model is shown in Figure 7a. Here, we use the high-speed results in M = 0.593 and M = 0.845. A viscous compressible flow with an SST k-ω turbulence model in second-order discretization are selected in CFD simulation. The generated mesh on the AGARD-B model is of unstructured type with 10-layer prism cells (the growth rate 1.2) to capture the boundary layer in the vicinity of the walls. It includes about 3.5 million computational cells based on half of the computational domain. The comparison of CFD and wind tunnel results are presented in Figure 7b–d. As we can see, there are excellent agreements between the CFD and experimental results in all aerodynamic coefficients, both in M = 0.593 and M = 0.845.

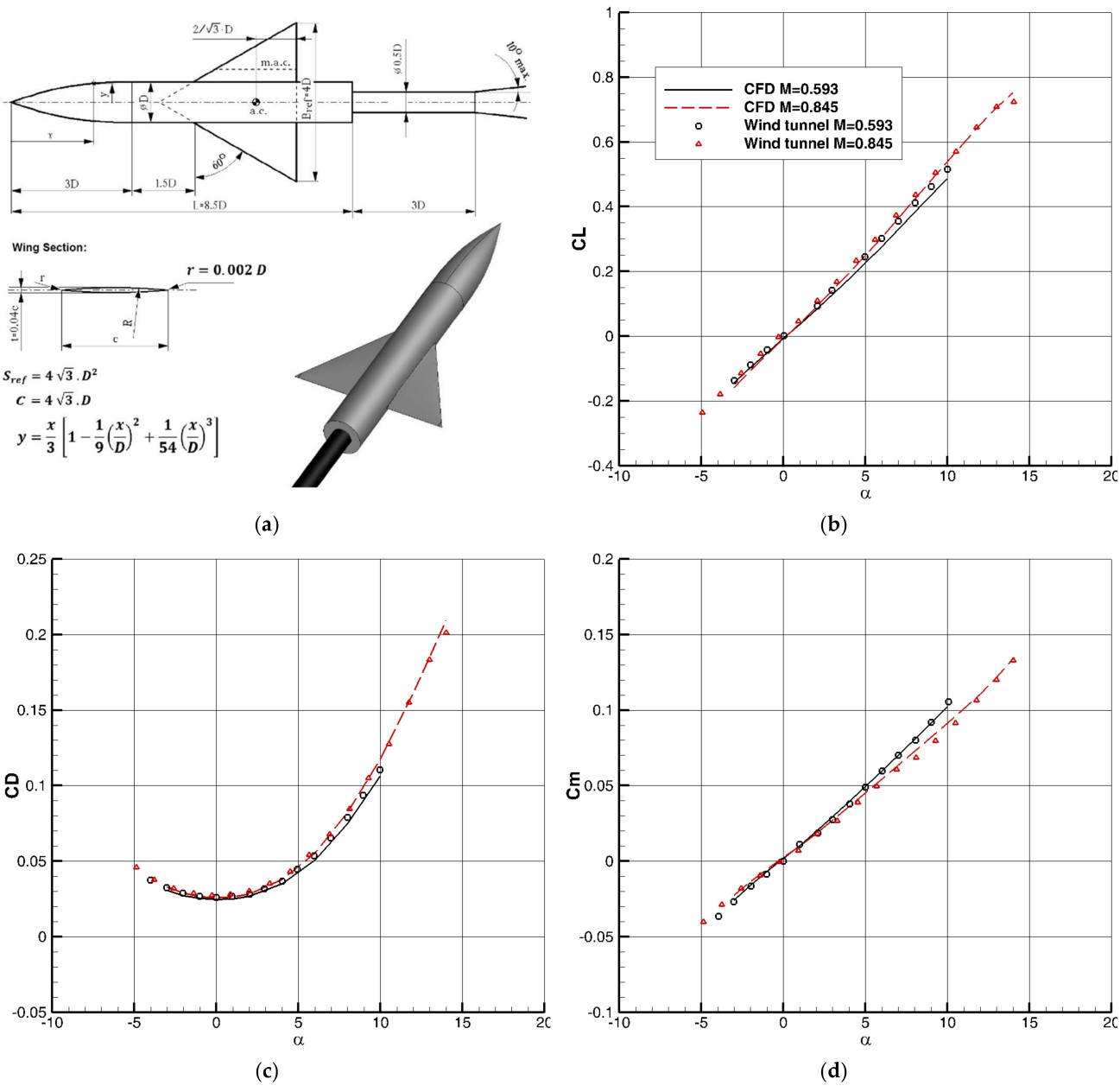

**Figure 7.** High-speed test case CFD validation: (**a**) AGARD-B model; (**b**) $C_L - \alpha$ diagram; (**c**) $C_D - \alpha$ diagram; (**d**) $C_m - \alpha$ diagram.

## 6. Results and Discussion

As mentioned before, the results of this research are based on numerical methods. Two test cases ensure the verifying of the solution, as well as the validating CFD method. Here, a mesh study is performed on the considered case in order to complete the validation of the numerical results. The generated mesh on the model is of unstructured type with 10-layer prism cells (the growth rate of 1.2) to capture the boundary layer in the vicinity of the walls. It includes about 7.2 million computational cells in fine case based on half of the computational domain. The results of the mesh study, including the representation of the boundary-layer mesh in the vicinity of the wing leading edge, are shown in Figure 8. The results show that the fine mesh is sufficient for numerical analysis.

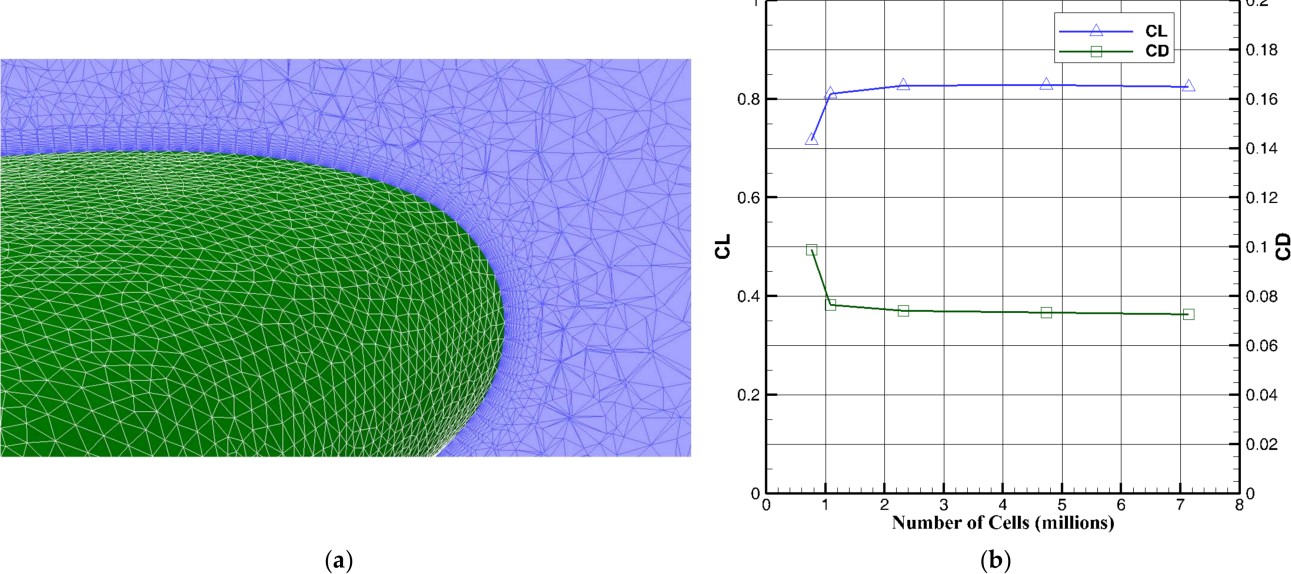

(**a**)                                                                                   (**b**)

**Figure 8.** Mesh study. (**a**) The generated mesh in the vicinity of wing leading edge; (**b**) the $C_L$ and $C_D$ mesh independency in the clean configuration at $\alpha = 10^\circ$.

### 6.1. Effect of PVG on the Clean Configuration

In the first step, the effects of PVG on a clean wing-body configuration of a low-wing aircraft in low-speed flow were investigated, which can simulate T.O and landing conditions. The results are presented in Figure 9 in airspeed velocity of V = 60 m/s. As we can see in the $C_L - \alpha$ diagram, by using PVG, the $C_{Lmax}$ of the wing-body increases from 1.12 to 1.42 due to the six-degree stall delay. On the contrary, at a high angle of attack, where the absence of PVG leads to a stall and, therefore, a sudden jump in drag, the use of PVG results in less drag. Figure 9c shows that PVG has no effect on the $C_m - \alpha$ slope and $C_{m0}$. This means that the installation of PVG has no effect on the stability of the aircraft, while stability is reduced in LEX configuration aircraft.

In Figure 9d, in the absence of PVG, at $\alpha = 19^\circ$, wide flow separation and reverse flow are observed on the wing. In Figure 8e, the flow path lines caused by the PVG vortex at the same angle of attack show how the presence of the PVG has caused the attached flow on the wing. Finally, at $\alpha = 20^\circ$, in Figure 9f, the vortex breakdown in the PVG leads to a wide separation in the wing. To elucidate the physics governing the flow, the spanwise distribution of the wing lift per unit of span segment length (Dy) is shown in Figure 10a. According to the results, the spanwise distribution of the lift force until the stall occurs in the case without PVG is almost the same, and only the presence of PVG has caused a slight decrease in the lift force in the area near the wing root. At higher angles of attack, the lift force distribution in the case of PVG still maintains its normal shape until the stall. Additionally, the component-based of $C_L - \alpha$ is presented in Figure 10b. As it can be seen, PVG has a small contribution (less than 1%) in the production of total lift. Hence,

the increase in the aircraft lift coefficient is due to the effect of the vortex produced by the PVG on the wing, not the PVG itself.

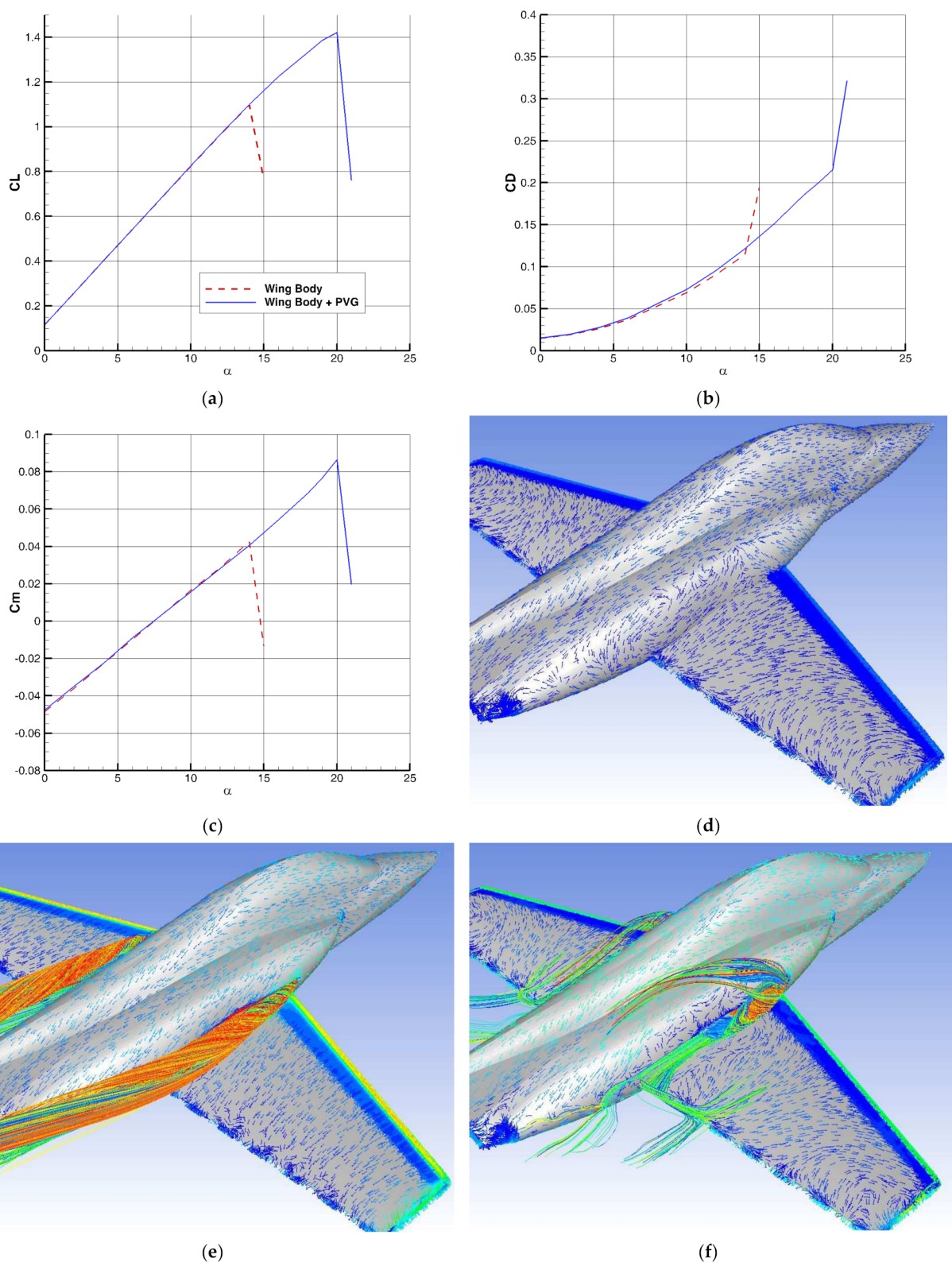

**Figure 9.** Effect of PVG on the clean configuration in V = 60 m/s: (**a**) $C_L - \alpha$ diagram; (**b**) $C_D - \alpha$ diagram; (**c**) $C_m - \alpha$ diagram; (**d**) the vectors at $\alpha = 19°$ in the absence of PVG; (**e**) the path lines of PVG and wing-body vectors at $\alpha = 19°$; (**f**) the PVG vortex breakdown in stall point at $\alpha = 21°$.

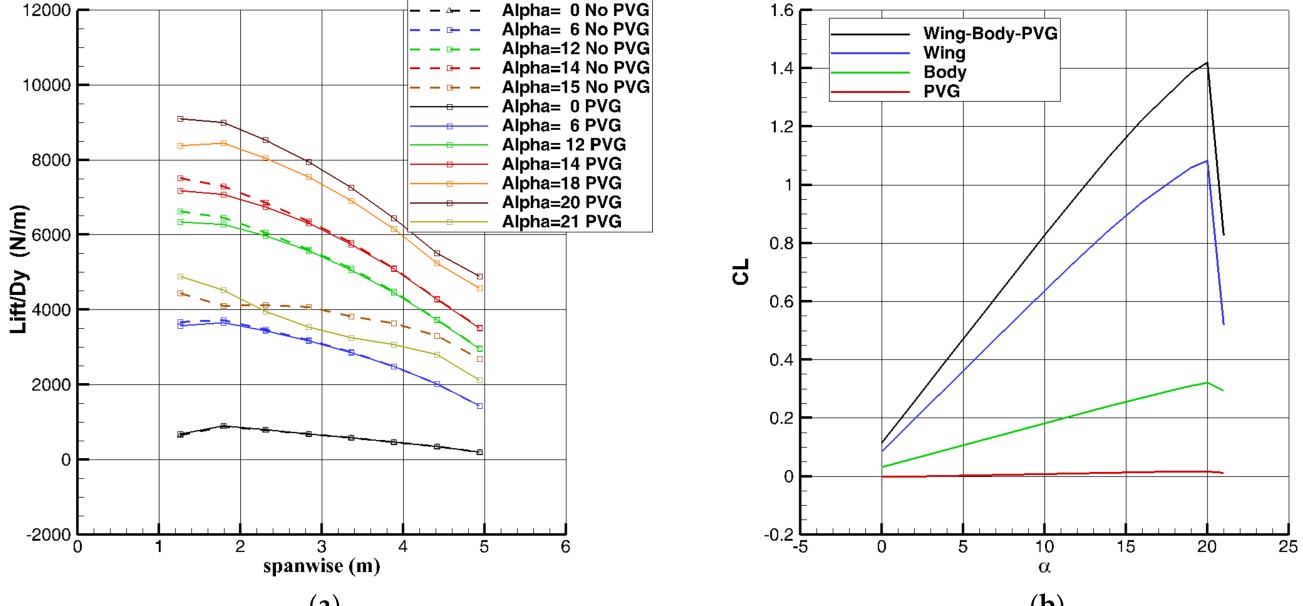

**Figure 10.** Effect of PVG on the clean configuration of the wing-body: (**a**) spanwise lift distribution; (**b**) component contribution in $C_L - \alpha$ diagram.

Now, our research focuses on high-speed flows. Two cases are more important here, one at the point of high-speed maneuvering and the other at MMO conditions. These two points, along with T.O and landing conditions, can show a general aerodynamic overview of an advanced training jet. M = 0.6, h = 18,000 ft., and $n = 4$ are selected as the high-speed maneuver point where $n$, the load factor of a maneuvering condition, is the ratio of lift to weight ($L/W$) of an aircraft (Equation (2)).

$$n = \frac{L}{W} = \frac{q.S}{W} C_L \qquad (2)$$

In Equation (2), $q$ and $S$ are dynamic pressure and wing reference area, respectively. Additionally, $C_L$ is the lift coefficient of the aircraft in the maneuver point.

M = 0.87 and h = 30,000 are chosen as an MMO condition. The results are presented in Figures 11 and 12. In the $C_L - \alpha$ diagram of M = 0.6 and h = 18,000 ft. (Figure 11a), *n = 4* is approximately at $C_L = 0.62$, at which point the results are close together. Additionally, at this point, at $C_L = 0.62$, $\alpha$ is about 6 degrees. At this angle, as well as almost most angles, referring to Figure 11b,c the drag and pitching moment coefficients are the same.

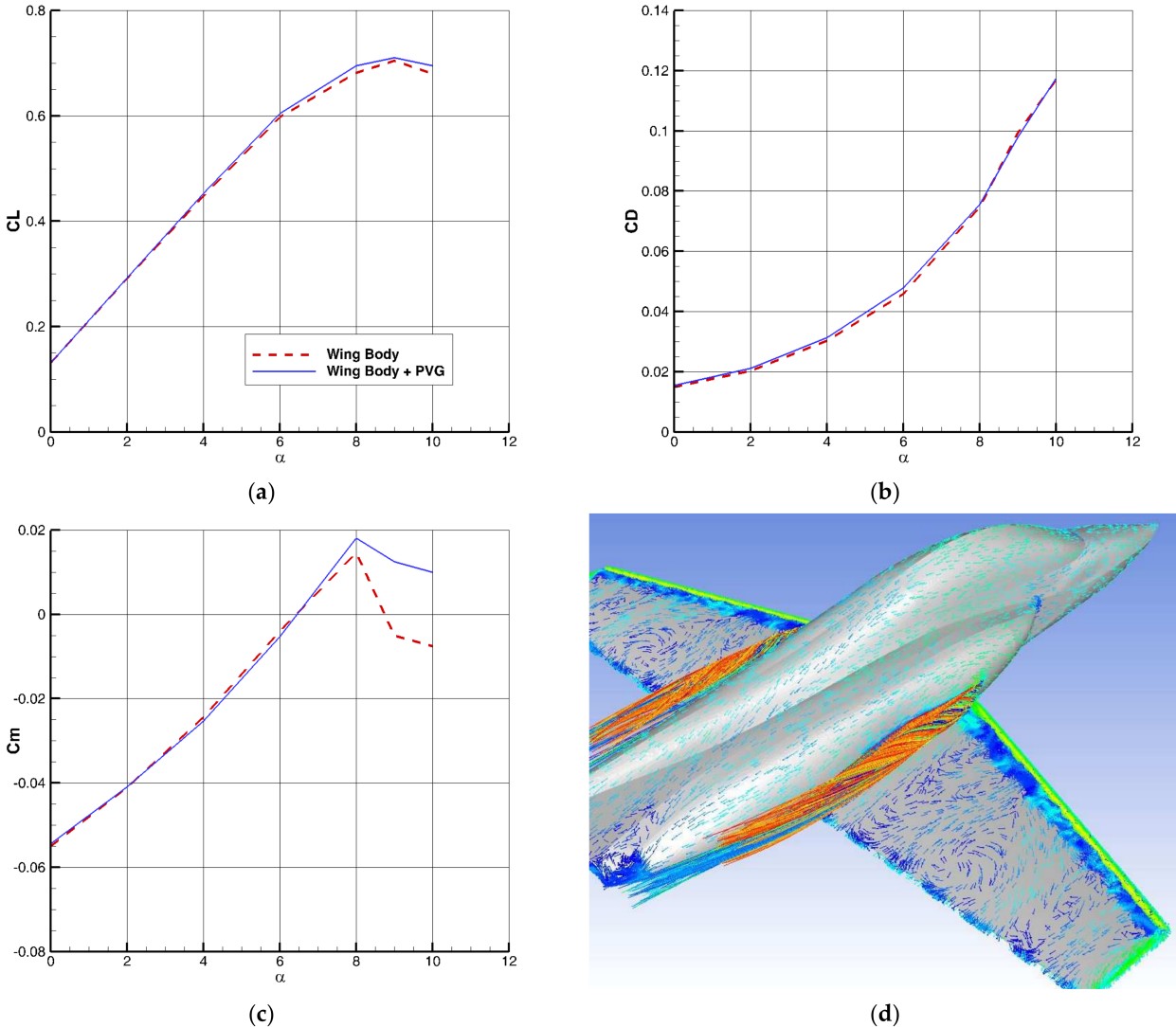

**Figure 11.** Effect of PVG on the clean configuration of the wing-body in M = 0.6, h = 18,000 ft: (**a**) $C_L - \alpha$ diagram; (**b**) $C_D - \alpha$ diagram; (**c**) $C_m - \alpha$ diagram; (**d**) the path lines of PVG and wing-body vectors at $\alpha = 9°$.

The path lines of PVG and wing-body vectors at $\alpha = 9°$ are shown in Figure 11d. In this angle of attack, the vortex power of PVG is not sufficient to prevent wing separations.

Additionally, in the MMO condition (M = 0.87, h = 30,000), the MMO cruise condition (*n* = 1) is approximately at $C_L = 0.13$, which corresponds approximately to $\alpha = 0°$ in the $C_L - \alpha$ diagram (Figure 12a). At this angle of attack, referring to Figure 12b,c, as well as almost most angles, the results are the same. The path lines of PVG and wing-body vectors at $\alpha = 8°$ are shown in Figure 12d. In this angle of attack, the vortex power of PVG is not sufficient to prevent wing separations.

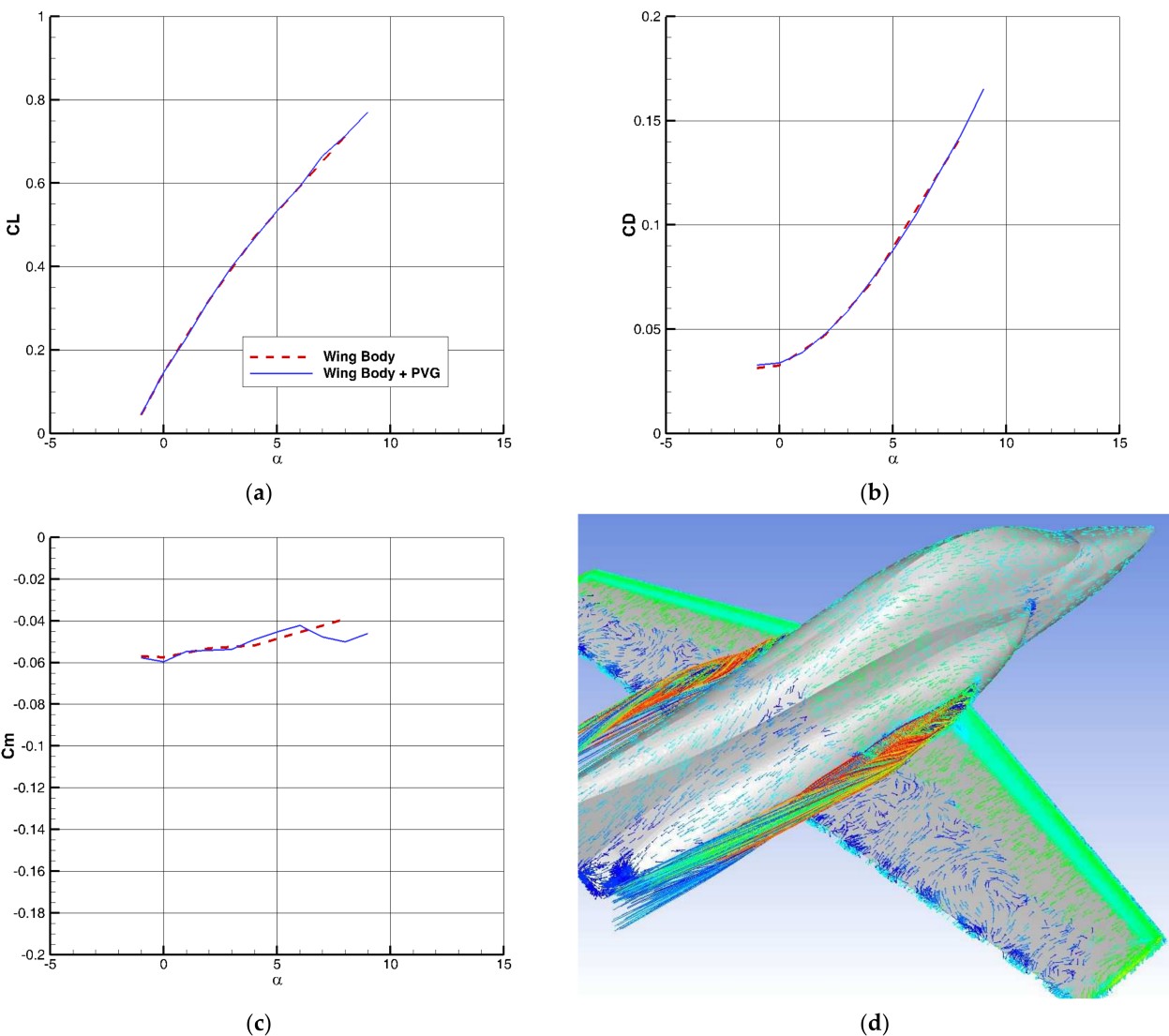

**Figure 12.** Effect of PVG on the clean configuration of the wing-body in M = 0.87, h = 30,000 ft: (**a**) $C_L - \alpha$ diagram; (**b**) $C_D - \alpha$ diagram; (**c**) $C_m - \alpha$ diagram; (**d**) the path lines of PVG and wing-body vectors at $\alpha = 8°$.

### 6.2. Optimization Results

A comparison of the optimized airfoil mentioned in Sec. 3 and the initial modified SC(2)-410 airfoil in the presence of PVG in both low-speed and high-speed flows are shown in Figures 13–15. As we can see in Figure 13a, in the $C_L - \alpha$ diagram, the $C_{Lmax}$ of the wing-body increases from 1.42 to 1.48 in optimized airfoil due to the 1.5-degree stall delay. In the $C_D - \alpha$ diagram, a slight decrease in $C_{D0}$ is observed in the optimized airfoil. In Figure 11c, the optimized airfoil causes a considerable upward shift in the pitching moment. Although an increase in pitching moment at high angles of attack increases the trim drag, in the flap-down position where there is a large negative pitching moment, this positive value reduces the trim drag.

In Figure 13d, in the presence of PVG at $\alpha = 21°$, the flow path lines caused by the PVG vortex are shown. The flow is attached to a wide region of the wing.

In the high-speed case of maneuvering point in the $C_L - \alpha$ diagram of M = 0.6 and h = 18,000 ft. (Figure 14a), $n = 4$ is approximately at $C_L = 0.62$, at which point the results are close together. At $C_L = 0.62$, $\alpha$ is about 6 degrees. At this angle, referring to Figure 10b, the drag coefficient is smaller in the optimized case. Also in the diagram, the optimized airfoil increases the stall delay by about 1 degree. In Figure 14c, an optimized

airfoil causes a considerable positive shift in the pitching moment coefficient. At $C_L = 0.62$, the absolute values of the pitching moment are close together.

In the MMO condition (M = 0.87, h = 30,000), the MMO cruise condition ($n = 1$) is approximately at $C_L = 0.13$, which corresponds approximately to $\alpha = 0^\circ$ in the $C_L - \alpha$ diagram in the initial airfoil, and $\alpha =$ in the optimized case (Figure 15). At these angles, in the case of the optimized airfoil, the drag coefficient is slightly lower, and the positive shift in the pitching moment will further reduce it due to reducing the trim drag. The reduced absolute value of pitching moment, in the case of the optimized airfoil, reduces the aft-body air-load on the aircraft.

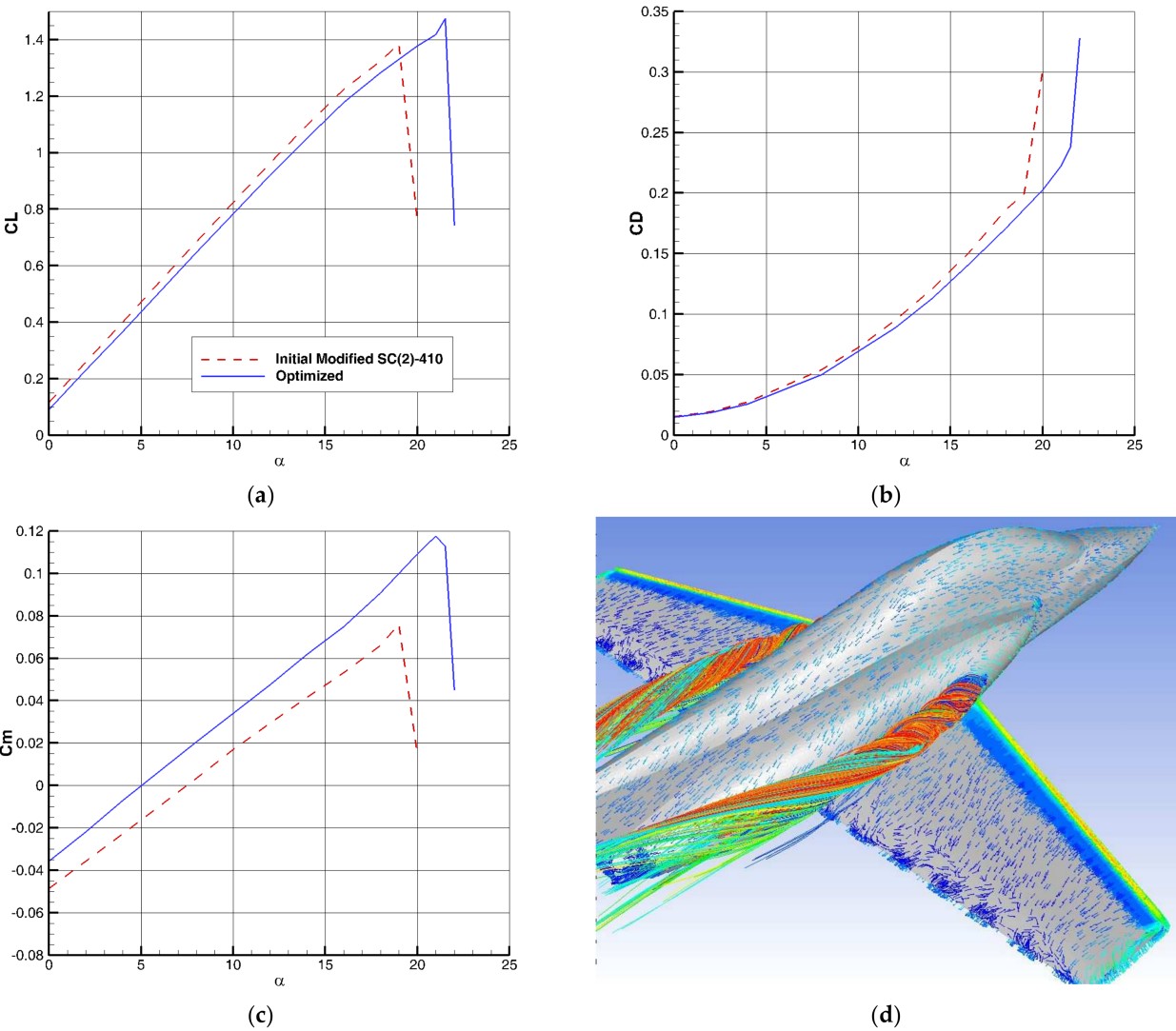

**Figure 13.** Comparison of the initial and optimized airfoil of wing-body in the presence of PVG in V = 60 m/s: (**a**) $C_L - \alpha$ diagram; (**b**) $C_D - \alpha$ diagram; (**c**) $C_m - \alpha$ diagram; (**d**) the path lines of PVG and wing-body vectors at $\alpha = 21^\circ$.

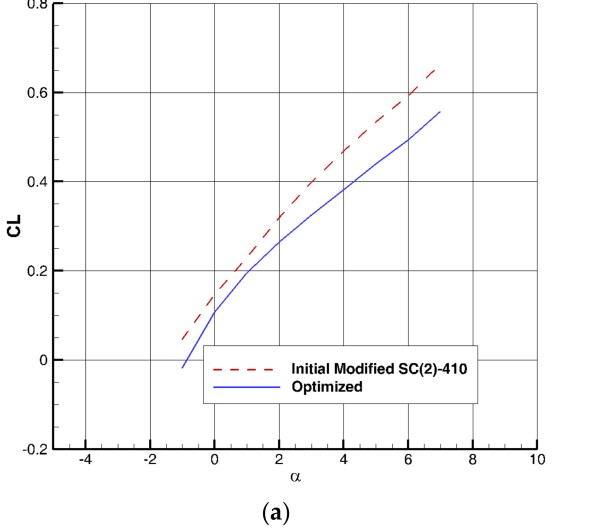

**Figure 14.** Comparison of the initial and optimized airfoil of wing-body in the presence of PVG in M = 0.6, h = 18,000 ft: (**a**) $C_L − \alpha$ diagram; (**b**) $C_D − \alpha$ diagram; (**c**) $C_m − \alpha$ diagram.

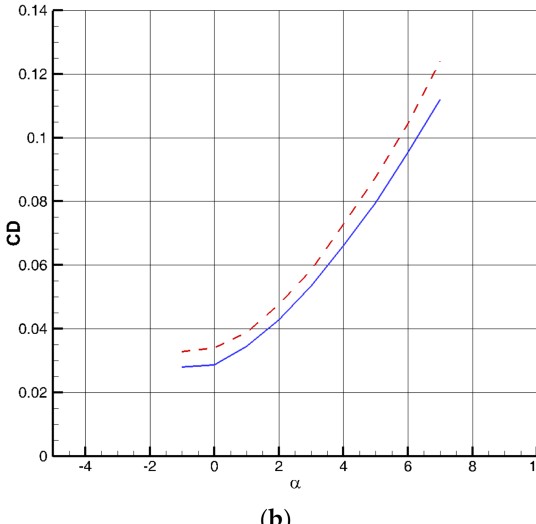

**Figure 15.** *Cont.*

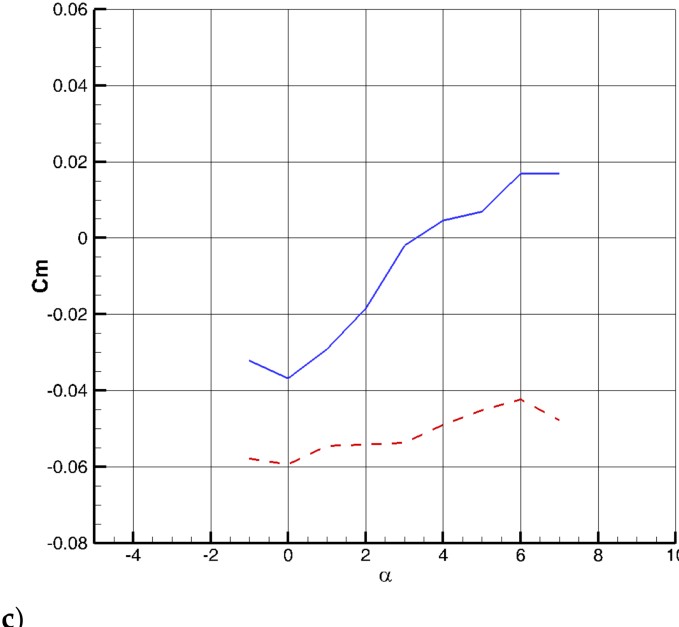

**(c)**

**Figure 15.** Comparison of the initial and optimized airfoil of wing-body in the presence of PVG in M = 0.87, h = 30,000 ft: (**a**) $C_L - \alpha$ diagram; (**b**) $C_D - \alpha$ diagram; (**c**) $C_m - \alpha$ diagram.

Since the results of PVG in the case of optimized airfoil show relatively better results, in the following, the effects of PVG on the flap-on configurations are investigated with the optimized airfoil. In other words, it can be noted that the combination of PVG as a fixed component high-lift device with a lifting-surface optimizing method, according to the categories in Figure 1, is suitable, and in the next step, this combination, as a passive method, is combined with a semi-passive method, i.e., movable flaps, again according to categories in Figure 1.

### 6.3. Hinged Leading Edge and Trailing Edge Flaps Configuration (HLF and HTF)

After examining the effects of PVG on the clean configuration of a wing-body of a low-wing type aircraft, we examine its effect on flap-down configurations. The results presented on the flaps are all related to the optimized airfoil. Here, we consider $HLF = 20^{\circ}$ and $HTF = 40^{\circ}$. The HLF extends over the entire leading edge of the wing, while the HTF covers part of it (Figure 16a). The results are shown in Figure 16b–d in V = 60 m/s. As we can see in the $C_L - \alpha$ diagram, $C_{Lmax}$ of the configuration increases from 1.21 to 1.80 due to the $12^{\circ}$ stall delay. In the $C_D - \alpha$ and $C_m - \alpha$ diagrams, the results are close to each other before high attack angles.

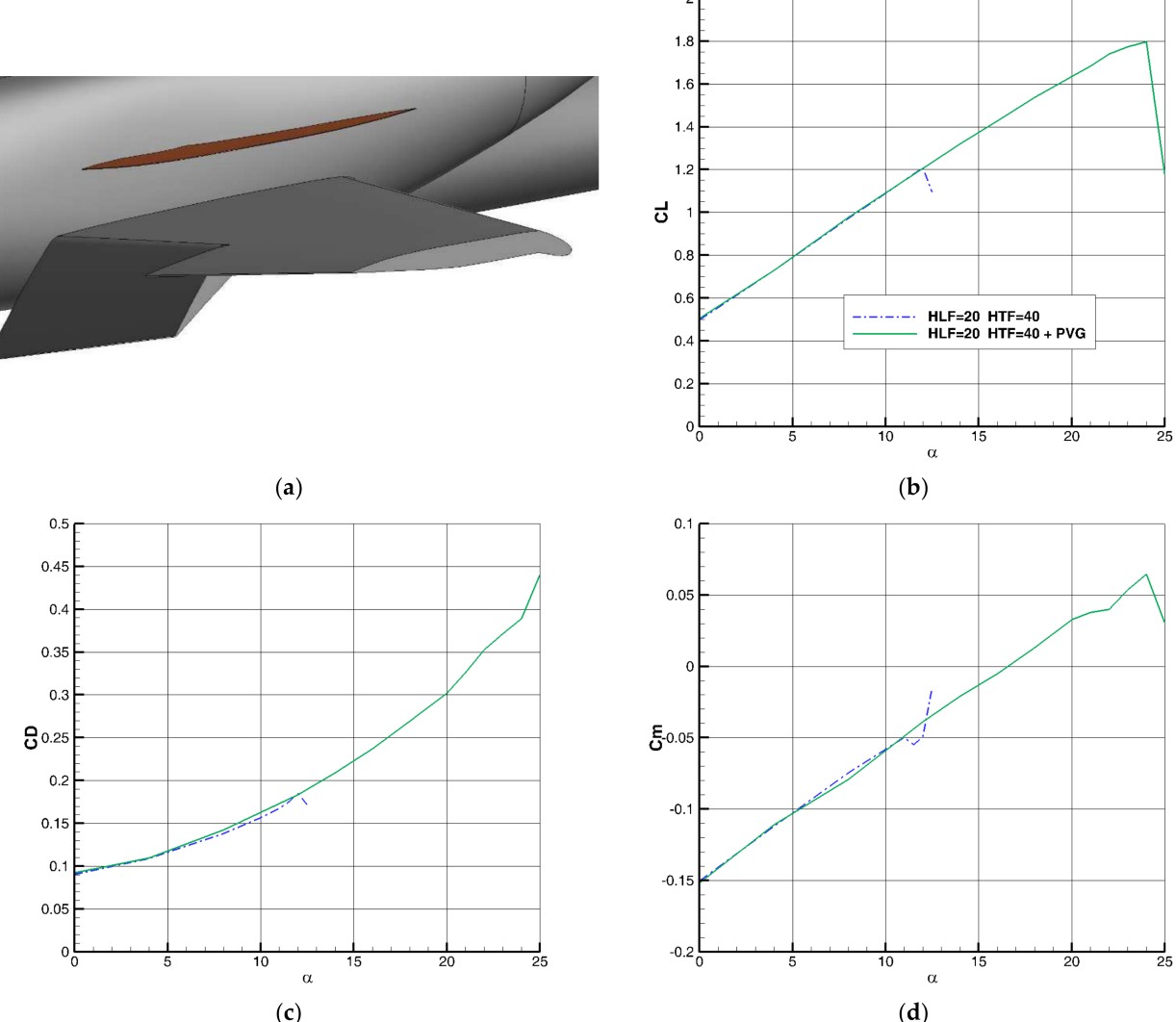

**Figure 16.** Effect of PVG on the HLF and HTF configurations of the wing-body in V = 60 m/s: (**a**) the HLF and HTF configurations; (**b**) $C_L - \alpha$ diagram; (**c**) $C_D - \alpha$ diagram; (**d**) $C_m - \alpha$ diagram.

*6.4. Hinged Leading-Edge Flap (HLF) and Single Slotted Trailing-Edge Flap (SSTF) Configurations*

The effect of PVG on some different combinations of HLF and SSTF configurations is investigated. Figure 17a shows the HLF-SSTF configuration. The results are shown in Figure 17b–d at V = 60 m/s. As we can see in the $C_L - \alpha$ diagram, $C_{Lmax}$ increases from 1.45 to 1.83 due to the $7^\circ$ stall delay at $SSTF = 30^\circ$ and from 1.52 to 1.91 due to the same stall delay in $SSTF = 40^\circ$. In the configuration $HLF = 20^\circ; SSTF = 40^\circ$, $C_{Lmax}$ reaches to 2.05 in $\alpha = 25^\circ$ in the presence of PVG instead of 1.4 in $\alpha = 12^\circ$ in its absence. In the $C_D - \alpha$ and $C_m - \alpha$ diagrams, the results in the corresponding configurations are almost close to each other before high angles of attack. Figure 18 shows the path lines generated by the PVG and wing vectors in HLF = 20 and SSTF = 40 at $\alpha = 25^\circ$. The strong generated vortex prevents the separations in the wing in the presence of the PVG (Figure 18a), while a wide flow separation leading to a large reversed flow on the wing is observed in the absence of PVG (Figure 18b).

**Figure 17.** Effect of PVG on the HLF and SSTF configurations of the wing-body in V = 60 m/s: (**a**) the HLF and SSTF configurations; (**b**) $C_L - \alpha$ diagram; (**c**) $C_D - \alpha$ diagram; (**d**) $C_m - \alpha$ diagram.

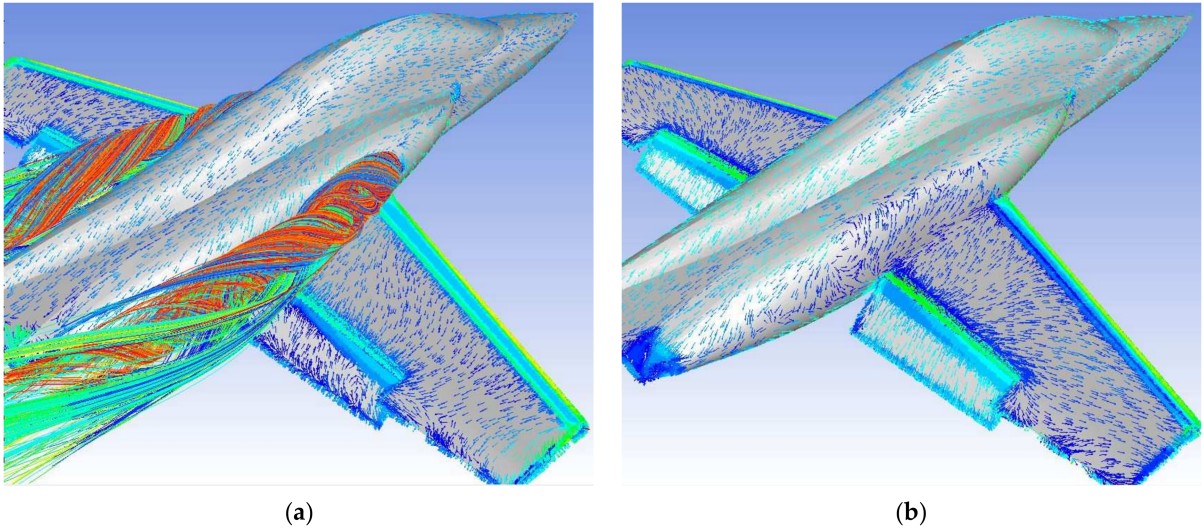

**Figure 18.** Vectors in the HLF and SSTF configurations of the wing-body in V = 60 m/s at $\alpha = 25°$: (**a**) in the presence of PVG and its path lines; (**b**) in the absence of PVG.

*6.5. Effect of Wing Twist on the PVG Effectiveness*

Since the PVG is located above the root of the wing, it seems that its power is greater in the inner region of the wing than in the outer region. Therefore, in wings with a negative twist angle that delays the stall at the tip of the wing, PVG should have a greater effect. There is a twist angle of $-2°$ in the wing in all cases in the results presented in previous sections. Now, we examine the effect of PVG in the case of no twisting of the wing. Figure 19 shows the results. In the case without PVG, the early separation of the flow in the wing, which started gradually from the angle of attack 8°, is delayed by $-2°$ of twist in the wing, which is a normal result. The comparison of two different twist angles in the presence of PVG shows that removing the wing twist has reduced the value of $C_{Lmax}$ from 1.48 to 1.39, and the stall angle has also occurred about $1.5°$ earlier. Therefore, to increase the effectiveness of PVG, it is better to have a wing with a negative twist angle.

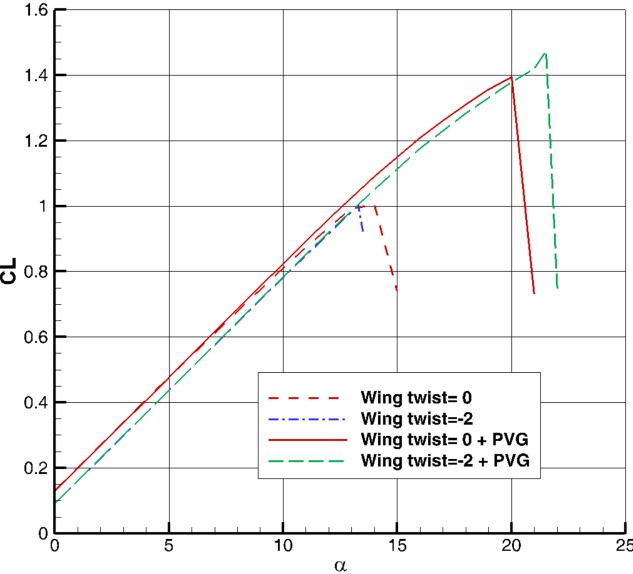

**Figure 19.** Effect of wing twist on the PVG effectiveness.

**7. Conclusions**

In this research, a new aerodynamic surface called PVG was introduced. Like the LEX and nacelle/body strake, this surface creates a strong vortex on the wing and delays the flow separation, but unlike the LEX, it does not change the stability and aerodynamic center of the aircraft. The results of this study were analyzed on a low-wing advanced training jet. In the clean configuration, the $C_{Lmax}$ of the wing-body increases from 1.12 to 1.42 due to the six-degree stall delay in the presence of the PVG. Additionally, PVG does not show an inappropriate effect on aerodynamic coefficients at high speeds.

The optimization of the lifting surface as a passive BLC, using the adjoint method, causes further improvement in the value of $C_{Lmax}$ so that this value increases from 1.42 to 1.48. The changes in the behavior of aerodynamic coefficients at high speeds are small.

In the case of the hinged flaps of the leading edge and trailing edge ($HLF = 20°$ and $HTF = 40°$), the value of $C_{Lmax}$ increased from 1.21 to 1.8. Additionally, a significant increase in the value of $C_{Lmax}$ is observed in the cases of single-slotted flaps of the trailing edge. In the case of $HLF = 20°$ and $SSTF = 40°$, the value of $C_{Lmax}$ increases from 1.4 to 2.05 due to $13°$ delay in the stall angle. According to the results, the efficiency of PVG is much higher than that of the nacelle strake. This is due to the proposed size and position of PVG, which makes it a powerful vortex generator. The new position also ensures that the aerodynamic center of the aircraft does not move.

One of the characteristics of PVG, which was identified by investigating the effect of wing twist on PVG efficiency, is that the vortex generated by this aerodynamic surface

affects the inner region of the wing more than its outer region. Therefore, the negative twist in the wing increases the effect of PVG.

According to the requirements of Figure 2, acceptance of a new device depends on various issues. The above-obtained results are in the fields of aerodynamics, performance, stability, control, and air-loading. They show that applying a PVG on a low-wing aircraft with a speed range up to high subsonic Mach numbers improves the performance of the aircraft at low speeds without improper effects at high speeds. Additionally, due to the low contribution of lift force related to the PVG itself, there is not much aerodynamic load on it. For example, the estimation of the presented aircraft at sea level in M = 0.87 shows that the maximum load on the PVG is about 110 kg. Therefore, the structure required for PVG is simple and light and is negligible compared to the weight of the aircraft. This simple structure does not have a considerable negative effect on the maintainability of the aircraft.

Since the PVG aerodynamic surface is introduced for the first time in this work, more research is needed to optimize its effects as well as its possible applications. The additional works are suggested for the future as follows:

- Complementary research on PVG effects in full aircraft configuration, including horizontal and vertical tails;
- Research on combining PVG with other passive devices such as VGs, dimples, fish scales, etc.;
- PVG shape optimization using the adjoint method or other optimization methods;
- Investigations on PVG with the variable angle of incidence;
- The possibility of use of PVG in low-wing business jets to increase T.O weight.

**Author Contributions:** M.G.: Conceptualization, methodology, software, validation, formal analysis, investigation, resources, data curation, writing—original draft preparation, writing—review and editing, visualization, supervision, and project administration. M.M.: validation, formal analysis, investigation, data curation, writing—review and editing, visualization, and project administration. All authors have read and agreed to the published version of the manuscript.

**Funding:** This research received no external funding.

**Data Availability Statement:** Not applicable.

**Conflicts of Interest:** The authors declare no conflict of interest.

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
