# Peer review of "Delaying the Stall of A Low-Wing Aircraft Using A Novel Powerful Vortex Generator"

_inventions, doi:10.3390/inventions7040095_

Round 1

Reviewer 1 Report

The Authors present a paper describing a novel type of high lift device they call a powerful vortex generator. Their work is presented as a case study where the new devices is fitted to a jet trainer type aircraft, and computational fluid dynamics is used to evaluate the changes to the flowfield. 

All in all, the scientific approach seems sound but the presentation could benefit from some minor additions/alterations.

1) A few plots of the spanwise lift-distribution on the main wing would give the reader a better understandning of wat is happening in the flowfield. How does the PVG influence the spanwise lift distribution? How much of the added lift is coming from the PVG:s themselves?

2)Fig 5a has jpeg compression artefacts in it and is hard to read.

3)In the intro: "intrinsic stability is a better choice..."  this is an opinion and need to phrased as such.

4) The similarity with nacelle strakes is obious, the intro should mention them and similarities/differences in function and flow influence.

Author Response

Please see atachment

Reviewer 2 Report

The article deals with the new method of passive flow control. The powerful vortex generator was used to change the aerodynamic properties of the tested model. The presented results are interesting but the method of investigation presentation is insufficient. Below is the list of my objections:

1. The introduction is poor and must be extended. The introduction presents only general, obvious knowledge and does not deal with different ways of passive control. The introduction must be revised and expanded.

2. The simulation method was described as insufficient. The mesh and models were not discussed, and the mesh parameters were not presented. The same with the simulation boundary layer. In which way was defined etc

3. The structure of the article is not clear and complicated. I suggest better describing what will be done in what order and for which parameters and model.

4. The parameters of the simulation were not clearly presented. I suggest adding a table in which the parameters of investigations will be presented.

5. Advantaged and disadvantages of PVG should be presented in the background of other methods.

6. The used aberration and symbols should be always described. Even if it is something so obvious like Cl.

7. I suggest changing the title of the article. The use of the aberration is senseless and unreasonable. The title suggests that the PVG can be used only in advanced low-wing training jets and you can underline that all aerodynamic parameters were changed by PVG. Suggest the use of the "novel" word. Please think about it.

Round 2

Reviewer 2 Report

The corrections made by the authors significantly increased the level of the paper. I have no objections.